# Evaluating miR-107 and adiponectin as biomarkers in obstructive sleep apnea: Associations with neurotransmitters and metabolic regulation

Asifa Ashraf[1,2], Muhammad Sarwar[1], Kaleem Arshad[2]*, Muhammad Ali Talat[2], Khudija Saleh[2]

1 The University of Lahore, Institute of Molecular Biology & Biotechnology, Lahore, Pakistan,
2 Biochemistry, Khawaja Muhammad Safdar Medical College, Sialkot, Pakistan

* kaleemarshad630@gmail.com

## Abstract

This study aimed to examine the levels of microRNA-107 (miR-107) and its correlation with neurotransmitters (glutamate, serotonin, melatonin) and adiponectin in patients with obstructive sleep apnea (OSA) compared to healthy controls. The results showed that serum levels of miR107, melatonin, and adiponectin were significantly lower in OSA patients compared to controls, while serotonin and glutamate levels were significantly higher. Spearman correlation analysis revealed that in the control group, miR-107 levels were moderately correlated with glutamate (negative) and adiponectin (positive), but these associations were disrupted in the OSA group. Receiver operating characteristic (ROC) curve analysis demonstrated that miR-107 had excellent diagnostic performance, with 100% sensitivity and 89.3% specificity at a cut-off of 3.0 ng/mL. Adiponectin also showed strong diagnostic potential, with 78% sensitivity and 89% specificity. In contrast, serotonin, melatonin, and glutamate exhibited more moderate diagnostic accuracy. These findings suggest that miR-107 and adiponectin could serve as promising biomarkers for diagnosing OSA, and targeting miR-107 to modulate metabolic factors may offer novel therapeutic approaches for improving OSA management.

## Introduction

Sleep is an intricate neurophysiological phenomenon characterized by an inherent predisposition to disengage from environmental stimuli, governed by a finely tuned interplay between sleep-promoting and arousal-regulating neural centers [1,2]. It serves a critical function in cerebral metabolic restoration, sensory attenuation, and cognitive consolidation [3]. OSA is primarily caused by disruptions in neural control of breathing and intermittent upper airway collapse, leading to inspiratory airflow limitations, sleep apnea, hypoxia, fragmented sleep, and arousals [4].

**Data availability statement:** All relevant data are within the paper and its Supporting Information files.

**Funding:** The author(s) received no specific funding for this work.

**Competing interests:** The authors have declared that no competing interests exist.

Different biomarkers have been used to assess the presence and severity of OSA and to evaluate its systemic effects. C-reactive protein (CRP), interleukins (IL-6, IL-8), and tumor necrosis factor-alpha (TNF-α) reflect systemic inflammation associated with OSA and its comorbidities, but they are not specific to OSA [5]. Oxidative stress biomarkers such as malondialdehyde (MDA), 8-isoprostane, and oxidized LDL are associated with intermittent hypoxia and reoxygenation, but they lack standardization [6]. Genetic biomarkers, such as TNF-α gene polymorphisms and angiotensin-converting enzyme gene polymorphisms, could potentially predict susceptibility to OSA or its complications. Genetic variability and environmental influences make it difficult to identify a single genetic marker for OSA [5]. The complex polygenic nature of OSA means that no single genetic marker is likely to be definitive. Current research is focused on identifying more precise, reliable biomarkers for OSA to complement polysomnography, the gold standard for diagnosis.

microRNA (miR), present in the cell as part of ribonucleoprotein complexes, plays a crucial role in conjunction with the RNA-induced silencing complex (RISC), where its RNA component serves as a probe facilitating RISC attachment to target DNA fragments, thereby modulating gene expression by inhibiting translation initiation or elongation, causing premature ribosomal dissociation from mRNA, and promoting degradation of nascent polypeptides [7,8]. The levels of miR-485-5p, miR-107, and miR-199-3p were downregulated in OSA patients compared to controls, while miR-574-5p showed upregulation in OSA patients compared to controls [9]. Additionally, miRs that affect sleep homeostasis and circadian rhythm include miR-107, miR-124, miR-125a-3p, miR-132, miR-182, miR-126, and miR-146a [10].

miR-107, a non-invasive and specific biomarker directly linked to the molecular events triggered by intermittent hypoxia, was selected for further study due to its strong correlation with the arousal index and significant association with sleep parameters [7]. It regulates key pathways involving glutamate metabolism, serotonin synthesis and signaling, and adiponectin expression. Dysregulation of miR-107 exacerbates glutamate toxicity, serotonin imbalance, and reduced adiponectin levels, contributing to cognitive impairment, respiratory dysfunction, and metabolic issues, particularly in obese individuals [11]. Targeting miR-107 could address these interconnected issues, potentially improving cognitive function, respiratory control, and metabolic health, thereby alleviating OSA severity. This rationale supports its selection as a promising biomarker for advancing the clinical management of OSA.

Glutamate, an excitatory neurotransmitter and a key metabolic intermediate in the brain, plays a critical role in the sleep-wake cycle, with its extracellular concentration tightly regulated by excitatory amino acid transporters, while glutamatergic neurons contribute to sleep facilitation under stressful conditions [12–14]. Elevated plasma glutamate levels have been associated with OSA-related health factors, including total and visceral obesity, dyslipidemia, insulin resistance, type 2 diabetes, and atherosclerosis, suggesting a distinct metabolomic profile that links OSA to cardiometabolic phenotypes [15,16].

Serotonin, a neurotransmitter and biogenic amine synthesized in the central and enteric nervous systems, is derived from the amino acid tryptophan [17,18].

Sleep-disordered breathing has been associated with lower blood serotonin levels and higher tryptophan hydroxylase 1 levels, with serotonin serving as a potent stimulator of central ventilation and maintaining upper airway patency through the chemoreceptor pathway; a relative reduction in serotonin contributes to the development and exacerbation of obstructive sleep apnea [19–21].

Melatonin, a hormone derived from serotonin in the pineal gland via the tryptophan-serotonin pathway, exhibits variable concentrations influenced by input from the brain's circadian centers and plays a key role in promoting sleep, modulating pituitary and adrenal hormones, regulating immune functions, and maintaining circadian rhythms [22–24]. OSA can trigger REM sleep behavior disorder (RBD) through apneic events linked to arousals, and treatment with 2mg prolonged-release melatonin has shown improvement in all patients with RBD, though it remains ineffective for untreated sleep-breathing disorders [25]. OSA significantly affects serum melatonin levels, and while short- or long-term CPAP treatment does not notably alter melatonin concentrations, a three-month CPAP regimen has been shown to restore the physiological rhythm of melatonin secretion in OSA patients [26].

Adiponectin, a protein hormone released from adipose tissue, possesses anti-hyperglycemic, anti-atherogenic, and anti-inflammatory properties, acting as a protective factor against obesity-related conditions [27–29]. It promotes fatty acid biosynthesis, inhibits hepatic gluconeogenesis, enhances glucose uptake in skeletal muscles, and exhibits antioxidant and anti-atherosclerotic effects [30,31]. A meta-analysis of 20 studies found that plasma adiponectin levels in patients with OSAS were significantly lower than those in age- and sex-matched controls, with similar findings observed in other investigations [32,33]. OSA, which can occur in both obese and non-obese individuals, is negatively associated with adiponectin and positively associated with cholesterol, with these associations being significant in men but not in women; the severity of OSA is generally less in non-obese individuals compared to those who are obese [34–36].

The aim of this study is to explore the potential use of miR-107, glutamate, serotonin, melatonin, and adiponectin as diagnostic biomarkers for OSA, and to investigate any correlation between miR-107 and other biomarkers, in order to enhance the understanding of the pathophysiological connections between these biomarkers, if any exist. Combining these biomarkers in a panel could improve the diagnosis, severity assessment, and treatment monitoring of OSA, and their mimics or inhibitors may be developed to aid in curing OSA. However, further research is needed to validate their specificity, sensitivity, and practicality in clinical settings, making this study an effort toward advancing this cause.

## Materials and methods

### Study design

This observational cross-sectional study aimed to assess the potential role of serum miR-107 levels as a diagnostic biomarker for patients with OSA. The research was conducted at Khawaja Muhammad Safdar Medical College and its affiliated Allama Iqbal Memorial Hospital, both in Sialkot, providing a diverse sample population and access to high-quality medical facilities and research equipment. Participant enrollment took place from April 1 to August 31, 2024. Informed verbal consent was obtained after explaining the research aims, objectives, and data handling procedures, with consent witnessed by the attending physician (pulmonologist). Participants were categorized into two groups based on the international classification of sleep disorders: cases (n = 70) and controls (n = 50). Exclusion criteria included the presence of diabetes mellitus, metabolic syndrome, neurological, psychiatric, or other medical disorders causing significant morbidity. Demographic details of the study participants are summarized in Fig 1. Blood samples were collected, centrifuged for 10 min at 2000 g, and serum samples were then stored at −80°C until use. This study was approved by the Research Ethics Committee of the Khawaja Muhammad Safdar Medical College Sialkot, ensuring adherence to ethical standards for research involving human participants. The procedures followed were in accordance with the ethical standards of the institution and with the Helsinki Declaration of 1975, as revised in 2013.

| Characteristics | Cases (n = 70) | Controls (n = 50) |
|---|---|---|
| Age (mean ± SD) | 45 ± 9 | 44 ± 8 |
| Gender distribution (n) | | |
| Female | 46 | 34 |
| Premenopausal | 13 | 27 |
| Postmenopausal | 33 | 07 |
| Male | 24 | 16 |
| BMI (mean ± SD) | 36.6 ± 8.6 | 27.3 ± 3.1 |

**Fig 1. Baseline characteristics of the case and control groups, including sample size (n), age, gender distribution, and body mass index (BMI).**

## Measurement of miR-107 and neurotransmitters levels by ELISA

The enzyme-linked immunosorbent assay (ELISA) kits were used to measure levels of miR-107, serotonin, melatonin, glutamate, and adiponectin according to the manufacturer's instructions. The kits included Human miR-107 ELISA kit (Shanghai Ideal Medical Technology, China), along with the following from Bioassay Technology Laboratory (Shanghai, China): Human Serotonin ELISA kit (E1128Hu), Human Melatonin ELISA kit (E1013Hu), Human Glutamate ELISA kit (E6750Hu), and Human Adiponectin ELISA kit (E1550Hu).

## Statistical analysis

Descriptive statistics, including mean, standard deviation (SD), standard error of the mean (SEM), kurtosis, and skewness, were calculated for all variables. A p-value of $< 0.05$ was considered statistically significant throughout the analysis. The Shapiro-Wilk test was employed to assess the normality of the data distribution. Non-parametric data (miR-107, serotonin, melatonin, glutamate, and adiponectin) were compared with the Mann-Whitney-U test. Correlation analyses between miR-107 and serotonin, melatonin, glutamate, and adiponectin were performed separately for the cases and controls using Spearman's rank correlation coefficient. ROC curves were generated by plotting sensitivity against 1-specificity to evaluate the diagnostic performance of the miR-107, serotonin, melatonin, glutamate, and adiponectin. Optimal cut-off value for the ROC curve was determined using the Youden Index (YI = sensitivity + specificity − 1).

## Results

Baseline characteristics for the case (n=70) and control (n=50) groups are summarized in Fig 1. The mean age was comparable between cases and controls, with cases averaging 45 ± 9 years and controls 44 ± 8 years. Gender distribution showed that, within the case group, there were 46 females (13 premenopausal, 33 postmenopausal) and 24 males, whereas the control group included 34 females (27 premenopausal, 7 postmenopausal) and 16 males. Body mass index (BMI) was notably higher in the case group (36.6 ± 8.6) than in the control group (27.3 ± 3.1), indicating a significant difference in baseline BMI between the groups.

Subgroup-specific descriptive statistics, including male and female cases and controls, as well as premenopausal and postmenopausal groups, are provided in the supplementary file (S2 File). However, these subgroup analyses should be interpreted with caution due to the limited number of participants in certain groups, particularly in the postmenopausal control group, as well as to some extent in the male and premenopausal groups.

Circulatory levels of miR-107, serotonin, melatonin, glutamate, and adiponectin, were measured in serum samples from both case and control groups using ELISA. Comparisons between groups revealed that serum miR-107, melatonin, and adiponectin levels were significantly lower in cases compared to controls. Conversely, serotonin and glutamate levels

were significantly elevated in cases relative to controls. These differences highlight potential biomarkers in serum that distinguish patients from healthy controls (Figs 2 and 3).Serum levels of miR-107, serotonin, melatonin, glutamate, and adiponectin in both cases and controls did not follow a normal distribution, as evidenced by the Shapiro-Wilk test (Fig 2).

Considering the non-normal distribution of the data, Mann-Whitney U tests were performed to compare the levels of these biomarkers between cases and controls. The analysis revealed significant differences in the levels of glutamate (U = 2725.5, p = 2.10× $10^{-7}$), adiponectin (U = 220, p = 3.89× $10^{-16}$), serotonin (U = 2987.5, p = 4.56× $10^{-11}$), and melatonin (U = 796, p = 3.86× $10^{-7}$). Notably, for miR-107, the test yielded a U value of 0, with a highly significant p-value of 1.24× $10^{-20}$. These findings indicate substantial differences in the distribution of these compounds between groups, suggesting potential associations with the studied condition.

Given the non-normal distribution of biomarkers, we employed the Spearman correlation test to examine relationships between miR-107 and other biomarkers within cases and controls separately. For the cases, the Spearman correlation analysis revealed negligible negative correlations between miR-107 and glutamate (-0.01, p = 0.9485), adiponectin (-0.06, p = 0.6225), and melatonin (-0.05, p = 0.6981). Additionally, a weak positive correlation was observed between miR-107 and serotonin (0.15, p = 0.217). However, none of these correlations reached statistical significance, indicating that variations in miR-107 levels are not significantly associated with changes in these biomarkers in cases. In contrast, the analysis of control samples indicated a statistically significant moderate negative correlation between miR-107 and glutamate (-0.37, p = 0.008), alongside a significant moderate positive correlation with adiponectin (0.29, p = 0.0429). Conversely, no significant associations were detected between miR-107 and serotonin or melatonin in the control group. Thus, while variations in miR-107 levels were significantly associated with glutamate and adiponectin levels in the control group, no significant associations were observed with serotonin or melatonin.

Additional subgroup analyses were conducted to compare miR-107 and other biomarkers across different groups, including male and female cases and controls, as well as premenopausal and postmenopausal cases and controls. No statistically significant correlations were observed, except for a significant correlation between miR-107 and serotonin in the male control group, and between miR-107 and glutamate in the female control group. Detailed results, including statistical values, are provided in the supplementary file (S2 File).

To assess the diagnostic performance of serum miR-107, serotonin, melatonin, glutamate, and adiponectin levels in distinguishing cases from controls, ROC curves were generated (Fig 4). Using a cut-off value of 3.0 ng/mL, miR-107 achieved a sensitivity of 100% and a specificity of 89.3% for detecting OSA in the case group relative to controls. The area under the ROC curve (AUC) for miR-107 was 94.7%, indicating excellent discrimination between cases and controls. For

| Variable | Mean | | SD | | SEM | | Kurtosis | | Skewness | | Shapiro-Wilk Statistic (p-value) | |
|---|---|---|---|---|---|---|---|---|---|---|---|---|
| | Cases | Controls | Cases | Controls | Cases | Controls | Cases | Controls | Cases | Controls | Cases | Controls |
| mRNA 107 (ng/ml) | 1.18 | 77.85 | 0.75 | 66.23 | .09 | 9.37 | 0.36 | 1.86 | 1.04 | 1.62 | 0.89 (2.1× $10^{-5}$) | 0.78 (3.2× $10^{-7}$) |
| Serotonin (ng/ml) | 213 | 44 | 221 | 29.11 | 26.47 | 4.11 | 2.19 | 2.38 | 1.69 | 1.33 | 0.77 (4.18× $10^{-9}$) | 0.88 (0.00018) |
| Melatonin (ng/L) | 33.58 | 105 | 40.26 | 83.4 | 4.81 | 11.79 | 1.84 | -0.43 | 1.51 | 0.63 | 0.81 (3.29× $10^{-8}$) | 0.91 (0.0016) |
| Glutamate (ng/ml) | 56.32 | 21.91 | 59.66 | 29.94 | 7.13 | 4.23 | 2.80 | 5.78 | 1.81 | 2.52 | 0.76 (2.15× $10^{-9}$) | 0.61 (3.1× $10^{-10}$) |
| Adiponectin (mg/L) | 4.12 | 27 | 5.28 | 21.12 | 0.63 | 2.99 | 4.24 | 0.09 | 2.11 | 1.04 | 0.72 (2.69× $10^{-10}$) | 0.86 (3.27× $10^{-5}$) |

**Fig 2. Descriptive statistics of biomarker levels in cases and controls.** This figure presents the mean, standard deviation (SD), standard error of the mean (SEM), kurtosis, skewness, and results from the Shapiro-Wilk test, including p-values, for each biomarker evaluated.

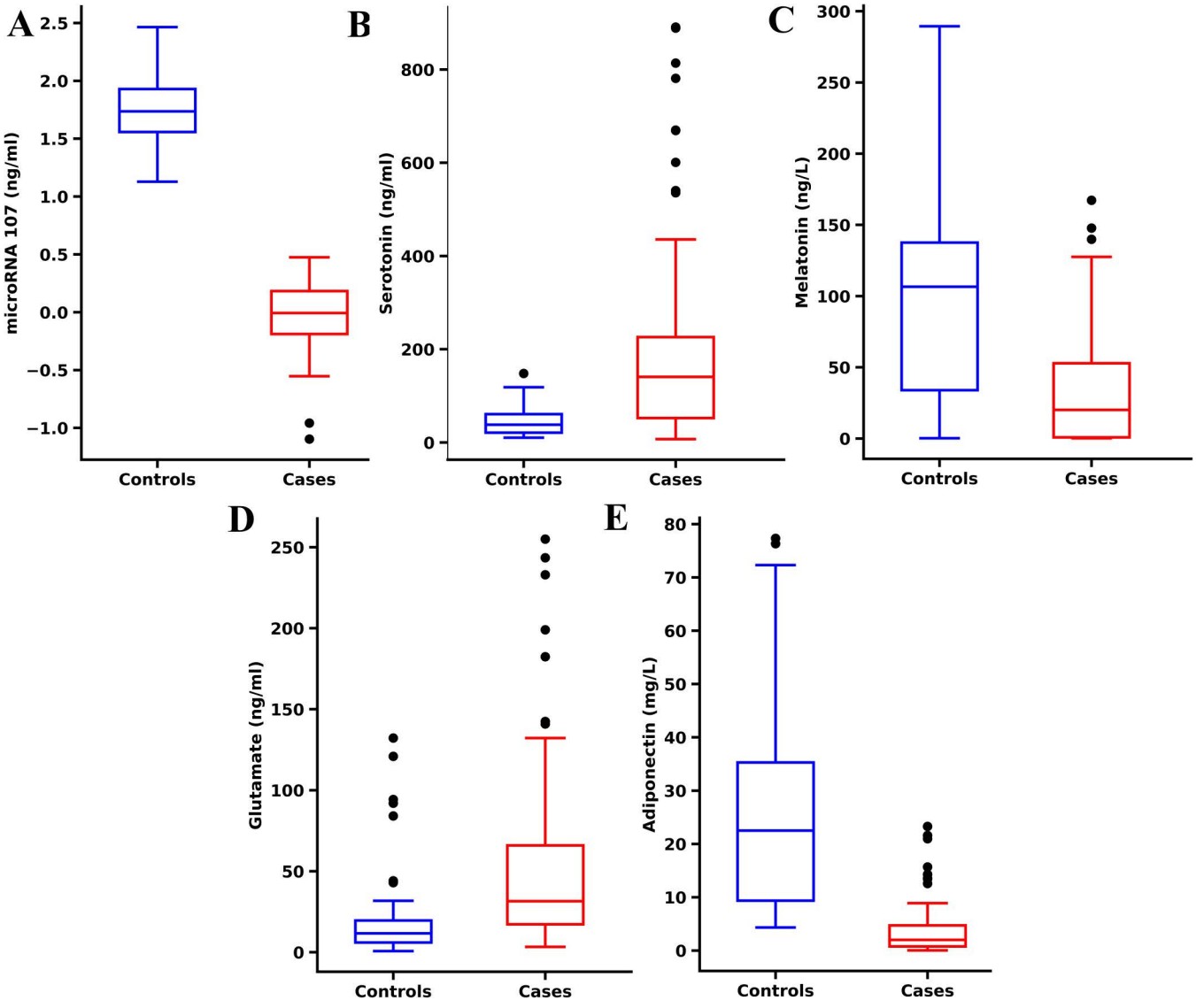

**Fig 3. The box and whisker plot (Tukey style, with outliers represented as black dots) illustrates the serum levels of (A) miR-107 (log-transformed), (B) serotonin (ng/mL), (C) melatonin (ng/L), (D) glutamate (ng/mL), and (E) adiponectin (mg/L).**

serotonin, a threshold of 125 ng/mL provided a sensitivity of 53% and a specificity of 94% for distinguishing cases from controls, with an AUC of 73.9%, indicating fair discriminatory capacity. The diagnostic assessment of melatonin levels, with a cut-off of 72 ng/L, yielded a sensitivity of 85.9% and a specificity of 55%, resulting in an AUC of 70.6%, reflecting moderate diagnostic accuracy. For glutamate, using a cut-off value of 22 ng/mL achieved a sensitivity of 65% and a specificity of 71%, with an AUC of 68.5%, indicating a modest distinction between the groups. Adiponectin, at a cut-off of 5 mg/L, demonstrated a sensitivity of 78% and a specificity of 89%, with an AUC of 83.7%, suggesting strong diagnostic potential. Overall, miR-107 and adiponectin exhibited high discriminatory power with AUCs of 94.7% and 83.7%, respectively, while serotonin, melatonin, and glutamate displayed moderate diagnostic utility in differentiating patients from controls.

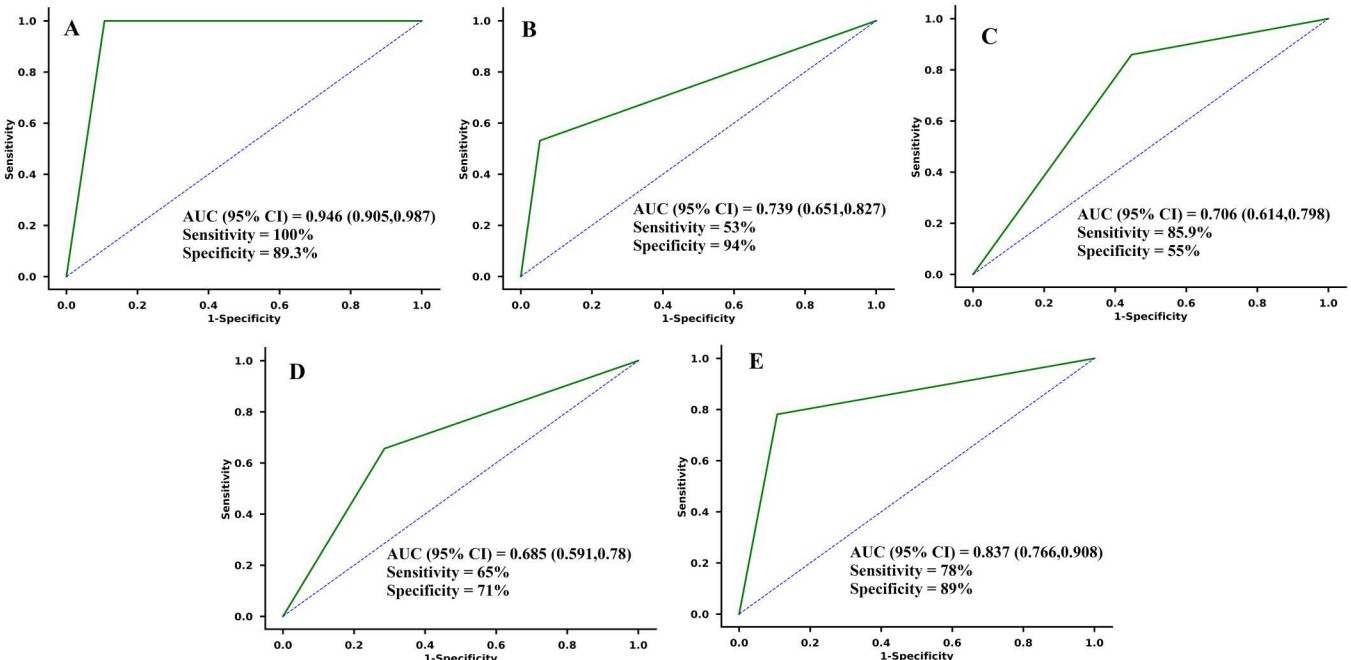

**Fig 4. Receiver operating characteristic (ROC) curve analysis to determine diagnostic performance of (A) miR-107 (AUC = 94.7%), (B) serotonin (AUC = 73.9%), (C) melatonin (AUC = 70.6%), (D) glutamate (AUC = 68.5%), and (E) adiponectin (AUC = 83.7%).**

## Discussion

The baseline characteristics of the case and control groups reveal significant demographic and health differences that may influence the study's outcomes. The mean ages of both groups were similar, with cases averaging 45 ± 9 years and controls at 44 ± 8 years, minimizing the impact of age as a confounding factor. However, notable differences were observed in gender distribution. The case group consisted of 46 females (13 premenopausal, 33 postmenopausal) and 24 males, while the control group had 34 females (27 premenopausal, 7 postmenopausal) and 16 males. The higher proportion of postmenopausal women in the case group could be significant, as menopause is often associated with metabolic and hormonal changes that may influence inflammation and other biomarkers. This discrepancy suggests that hormonal factors could play a role in the observed differences between groups, warranting consideration in further analysis.

A significant difference was observed in body mass index (BMI), with cases showing a higher mean BMI (36.6 ± 8.6) compared to controls (27.3 ± 3.1). This finding suggests that the case group is more likely to be overweight or obese, a factor known to influence metabolic processes and inflammatory states. Although participants with diabetes, metabolic syndrome, neurological, or other significant comorbidities were excluded, elevated BMI may still impact factors like lipid metabolism and systemic inflammation. Thus, BMI differences should be considered when interpreting biomarker results, as they may partly explain the variations observed between groups.

The serum levels of miR-107, serotonin, melatonin, glutamate, and adiponectin displayed distinct patterns between the case and control groups, suggesting their potential as biomarkers for differentiating the two. Specifically, miR-107, melatonin, and adiponectin levels were significantly lower in cases compared to controls, while serotonin and glutamate levels were higher in cases. These variations point to the possible involvement of these biomarkers in the condition under study, with altered levels reflecting underlying biological processes or disease mechanisms. Due to the non-normal distribution of these biomarker levels, confirmed by the Shapiro-Wilk test, the Mann-Whitney U test was applied to assess differences,

revealing statistically significant disparities for all markers. The extremely significant p-values, particularly for miR-107 (U = 0, p = 1.24 × 10^{-20}), further support the potential of these biomarkers in distinguishing between cases and controls.

Spearman correlation analysis was also conducted to explore the relationships between miR-107 and the other biomarkers within each group. In the case group, only weak and statistically insignificant correlations were found, suggesting that miR-107 levels did not meaningfully correlate with glutamate, adiponectin, melatonin, or serotonin. In contrast, moderate correlations were observed in the control group, with a statistically significant negative correlation between miR-107 and glutamate and a positive correlation with adiponectin. These findings suggest that, in the absence of the condition, miR-107 may be associated with glutamate and adiponectin levels. However, these associations appear disrupted or masked in the presence of the condition. This disruption could reflect disease-related changes in the regulatory mechanisms involving these biomarkers, further supporting their potential use in understanding and monitoring disease states.

The ROC has been analyzed for assessing the diagnostic performance of serum biomarkers. miR-107 demonstrated excellent discriminatory power, with an AUC of 94.7%, achieving perfect sensitivity (100%) and high specificity (89.3%) at a cut-off of 3.0 ng/mL. This suggests that miR-107 is a highly effective biomarker for diagnosing OSA. Adiponectin also showed strong diagnostic potential, with an AUC of 83.7%, demonstrating a sensitivity of 78% and specificity of 89% at a threshold of 5 mg/L. These results emphasize the high utility of miR-107 and adiponectin in distinguishing between cases and controls with strong accuracy.

In contrast, serotonin, melatonin, and glutamate demonstrated more moderate diagnostic utility. Serotonin, with an AUC of 73.9% at a cut-off of 125 ng/mL, showed fair discriminatory capacity, with 53% sensitivity and 94% specificity. Melatonin and glutamate exhibited moderate diagnostic accuracy, with AUCs of 70.6% and 68.5%, respectively. Melatonin had a sensitivity of 85.9% but a lower specificity (55%) at a cut-off of 72 ng/L, while glutamate's diagnostic accuracy was modest, showing 65% sensitivity and 71% specificity at a cut-off of 22 ng/mL. Overall, miR-107 and adiponectin emerged as the most promising biomarkers in this study, while serotonin, melatonin, and glutamate offered moderate potential for distinguishing between patients and controls.

Addressing high BMI through weight management is a key therapeutic strategy for OSA patients. Weight reduction can significantly alleviate OSA symptoms, improve breathing during sleep and reduce the risk of complications. miR-107 can serve as biomarker for metabolic diseases and may also be therapeutic targets for modulating neurotransmitters and hormonal levels aiming to restore metabolic balance. Although serotonin, melatonin and glutamate measurements alone may not provide definitive diagnosis they provide valuable insights in helping to identify, monitor and better understand OSA. Together miR-107 and adiponectin offer a promising diagnostic toolset. Their combined use may enhance early detection, track disease progression and support personalized treatment strategies making them valuable in the advancement of metabolic and cardiovascular disease diagnostics.

## Supporting information

**S1 File. Data of cases and controls, including miR-107, serotonin, glutamate, adiponectin, and melatonin levels measured by ELISA.**
(XLSX)

**S2 File. Detailed statistical analysis of various subgroups, including male and female cases and controls, as well as premenopausal and postmenopausal groups, covering descriptive statistics and correlation analyses.**
(XLSX)

## Author contributions

**Conceptualization:** Asifa Ashraf, Muhammad Sarwar.

**Formal analysis:** Asifa Ashraf, Kaleem Arshad, Muhammad Ali Talat, Khudija Saleh.

**Investigation:** Asifa Ashraf, Kaleem Arshad, Muhammad Ali Talat, Khudija Saleh.

**Methodology:** Asifa Ashraf, Muhammad Sarwar, Kaleem Arshad.

**Project administration:** Asifa Ashraf, Muhammad Sarwar.

**Resources:** Asifa Ashraf, Kaleem Arshad, Muhammad Ali Talat.

**Supervision:** Muhammad Sarwar, Khudija Saleh.

**Validation:** Asifa Ashraf, Kaleem Arshad, Muhammad Ali Talat.

**Visualization:** Kaleem Arshad.

**Writing – original draft:** Asifa Ashraf, Kaleem Arshad, Muhammad Ali Talat, Khudija Saleh.

**Writing – review & editing:** Asifa Ashraf, Muhammad Sarwar, Kaleem Arshad, Muhammad Ali Talat, Khudija Saleh.

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
