## [Decision Letter · Decision Letter 0]

2 Jan 2025

PONE-D-24-51520Evaluating microRNA 107 and adiponectin as biomarkers in obstructive sleep apnea: Associations with neurotransmitters and metabolic regulationPLOS ONE

Dear Dr. Arshad,

Thank you for submitting your manuscript to PLOS ONE. After careful consideration, we feel that it has merit but does not fully meet PLOS ONE’s publication criteria as it currently stands. Therefore, we invite you to submit a revised version of the manuscript that addresses the points raised during the review process. Please submit your revised manuscript by Feb 16 2025 11:59PM. If you will need more time than this to complete your revisions, please reply to this message or contact the journal office at plosone@plos.org . Please include the following items when submitting your revised manuscript:

We look forward to receiving your revised manuscript.

Kind regards,

Phakkharawat Sittiprapaporn, Ph.D.

Academic Editor

PLOS ONE

Journal Requirements:

2. In the ethics statement in the Methods, you have specified that verbal consent was obtained. Please provide additional details regarding how this consent was documented and witnessed, and state whether this was approved by the IRB.

3. We note that your Data Availability Statement is currently as follows: “All relevant data are within the manuscript and its Supporting Information files.”

Please confirm at this time whether or not your submission contains all raw data required to replicate the results of your study. Authors must share the “minimal data set” for their submission. PLOS defines the minimal data set to consist of the data required to replicate all study findings reported in the article, as well as related metadata and methods (https://journals.plos.org/plosone/s/data-availability#loc-minimal-data-set-definition ).

If your submission does not contain these data, please either upload them as Supporting Information files or deposit them to a stable, public repository and provide us with the relevant URLs, DOIs, or accession numbers. For a list of recommended repositories, please see https://journals.plos.org/plosone/s/recommended-repositories .

Reviewers' comments:

Reviewer's Responses to Questions

**Comments to the Author**

1. Is the manuscript technically sound, and do the data support the conclusions?

Reviewer #1: Yes

Reviewer #2: Yes

Reviewer #3: Yes

2. Has the statistical analysis been performed appropriately and rigorously? 

Reviewer #1: Yes

Reviewer #2: Yes

Reviewer #3: Yes

3. Have the authors made all data underlying the findings in their manuscript fully available?

Reviewer #1: Yes

Reviewer #2: Yes

Reviewer #3: Yes

4. Is the manuscript presented in an intelligible fashion and written in standard English?

Reviewer #1: Yes

Reviewer #2: Yes

Reviewer #3: Yes

5. Review Comments to the Author

Reviewer #1: The manuscript is understandable in general. The data and the conclusion are largely convincing. But there are some suggestion below.

Firstly, the comparison of miRNA 107 and neurotransmitters should be done as the following:

1.The comparisons between the female cases and controls and between the male cases and controls should be added.

2.Furthermore, the comparison between the premenopausal cases and controls as well as the comparison between the postmenopausal cases and controls should be added.

Secondly, the grammar and language need further revision. Since there are a lot of problems, here I just pointed out some issues but not all in this manuscript.

1.The full spelling for the abbreviation of ROC should be only addressed at the first time when it occurred in this manuscript but not each time later.

2.The full spelling of BMI should be presented at the first time when it occurred in this manuscript which is in the page 6th.

3. The last sentence in the page 4th is the same as the last sentence in the page 5th. There must be a mistake.

4.‘It acts as a protective endocrine that helps to prevent the development and progression of obesity related fatal conditions.’

In this sentence, ‘a protective endocrine’ should be ‘a protective endocrine factor’.

5.It has been found in a meta-analysis which included data until January 2018 which was based on 20 articles reported that the plasma levels of adiponectin in patients with OSAS were significantly lower than age/sex matched controls .

6.‘Lu et al has revealed that plasma adiponectin levels in OSA were lower that of control.’

This sentence should be revised as the following: Lu et al have revealed that the plasma adiponectin level in OSA were lower than that of control.

7.These association were significant for men but not for women.

In this sentence,‘were’ should be ‘was’.

8.‘adipose tissues’ should be ‘adipose tissue’.

9. ‘It has been found in a meta-analysis which included data until January 2018 which was based on 20 articles reported that the plasma levels of adiponectin in patients with OSAS were significantly lower than age/sex matched controls.'

This sentence can be revised as the following: It has been found in a meta-analysis, which included data up until January 2018 and was based on 20 articles, that the plasma levels of adiponectin in patients with OSAS were significantly lower than those in age/sex-matched controls.

10.These markers were independent of sleep duration; sleep fragmentation, insomnia and daytime sleepiness.

In this sentence, the ‘;’should be ‘,’.

11.‘These findings suggest that, in the absence of the condition, miRNA 107 may be associated with glutamate and adiponectin levels, but these associations appear disrupted or masked in the presence of the condition.’

In order to make this sentence more fluent, it can be revised as the following: These findings suggest that, in the absence of the condition, miRNA 107 may be associated with glutamate and adiponectin levels. However, these associations appear disrupted or masked in the presence of the condition.

Reviewer #2: Experiments have been conducted rigorously, with appropriate controls and replication. Sample sizes large enough to produce robust results. Methods and reagents are described in sufficient detail for another researcher to reproduce the experiments described.

The data presented in the manuscript support the conclusions drawn. The interpretation of results is justified and appropriate. Authors have discussed possible implications for their results. The article is presented in an intelligible fashion and is written in Standard English (clear, correct, and unambiguous).The article adheres to appropriate reporting guidelines and community standards for data availability. Results are appropriately reported.

Reviewer #3: In this manuscript, Asifa Ashraf et al. demonstrated the potential of miR-107 and Adiponectin as biomarkers for OSA. Overall, this is in several respects an interesting manuscript. I recommend for publication with some revisions.

1. Normally, miRNA detection is performed using qRT-PCR. The absolute content in serum needs to be measured, and a standard curve with miRNA standards should be used for calculation. However, the detection of miR-107 in this manuscript is described as using ELISA (Shanghai Ideal Medical Technology, China). Please clarify this method.

2. The notation of microRNA 107 is inconsistent. Generally, miR-107 would be a more appropriate format.

3. Considering that this study primarily demonstrates the potential of miR-107 and other markers as biomarkers for OSA, the authors should be cautious in describing many of the conclusions. For example, the last sentence of the abstract, “Understanding the relationship between … novel therapeutic approaches …” is not supported by the current findings.

4. I do not see a correlation analysis, especially between miR-107 and the biomarkers tested. Please clarify the significance of this analysis for the study, and describe it in an appropriate section.

5. The authors should enhance the logical flow of the Introduction. I believe it is more important to address existing biomarkers for OSA and their limitations, rather than listing the biomarkers the authors plan to investigate. Additionally, the authors mention other miRNAs besides miR-107 in the Introduction—why was miR-107 chosen for this study, and not the others?

6. Fig. 1 is a table.

7. The mean BMI of OSA patients is 36.6, while for the control group it is 27.3. Please discuss this in the Discussion section. Is the elevation of miR-107 and Adiponectin due to obesity or OSA? It is well known that Adiponectin is secreted by fat. Shouldn’t the ideal control group consist of subjects who are obese but do not have OSA?

6. PLOS authors have the option to publish the peer review history of their article (what does this mean? ). If published, this will include your full peer review and any attached files.

**Do you want your identity to be public for this peer review?** For information about this choice, including consent withdrawal, please see our Privacy Policy .

Reviewer #1: No

Reviewer #2: No

Reviewer #3: No

---

## [Author Response · Author response to Decision Letter 1]

25 Jan 2025

Phakkharawat Sittiprapaporn

Academic Editor

PLOS ONE

Dear Phakkharawat,

We would like to express our sincere gratitude for the opportunity to submit our manuscript titled " Evaluating miR-107 and adiponectin as biomarkers in obstructive sleep apnea: Associations with neurotransmitters and metabolic regulation" to PLOS ONE. We deeply appreciate the insightful comments and suggestions provided by you and the reviewers, which have greatly contributed to improving our work.

We have carefully considered each comment and made the necessary revisions to the manuscript. Below is our point-by-point response to the reviewers' comments:

Reviewer #1:

Comment 1: Comparison of miR-107 and Neurotransmitters Across Different Groups

We appreciate the reviewer’s insightful comments and suggestions. In response, we have conducted the requested analyses across the specified groups. The results of these analyses have been compiled in the supplementary file titled "Detailed Analysis." This file provides a comprehensive summary, including mean, standard deviation (SD), skewness, kurtosis, Shapiro-Wilk test statistics with corresponding p-values, Spearman's correlation coefficients with p-values, and Mann-Whitney U test statistics with p-values.

Our findings indicate that no statistically significant correlations were observed between miR-107 and other markers across all groups, except for significant correlations between miR-107 and serotonin in the male control group, and between miR-107 and glutamate in the female control group. Additionally, significant distribution differences were observed among all groups for all studied markers.

It is important to note that the initial analysis did not include these group-wise comparisons due to the limited number of participants in certain subgroups, particularly in the postmenopausal control group, as well as to some extent in the male and premenopausal groups. However, the overall sample size for the case and control groups was statistically sufficient to draw meaningful conclusions.

For future studies, we recommend ensuring adequate representation across all subgroups to facilitate more robust subgroup analyses. The primary objective of our study was to assess the feasibility of using these biomarkers for the diagnosis of OSA. We acknowledge this limitation and appreciate the reviewer’s valuable feedback, which has provided us with further insights to improve both the current study and our future research endeavors.

Thank you once again for your constructive input.

Comment 2: Manuscript Language and Clarity

We sincerely appreciate the reviewer’s valuable feedback and constructive suggestions. In response, we have thoroughly revised the entire manuscript, paying close attention to the issues highlighted. Additionally, we carefully reviewed the text to identify and correct any other potential errors. We are grateful for the reviewer’s efforts, which have significantly contributed to enhancing the quality and value of our article.

Reviewer #2:

We sincerely thank the reviewer for their positive feedback and appreciation of our work. We are pleased to know that our experimental rigor, methodological clarity, and data interpretation meet the required standards. Your encouraging comments motivate us to further uphold the quality and reproducibility of our research. We are grateful for your time and effort in reviewing our manuscript.

Reviewer #3:

Comment 1: Clarification of miR-107 Detection Method Using ELISA

We appreciate the reviewer’s concern. Due to the unavailability of qRT-PCR facilities at the time of our study, we opted for the ELISA method as a reliable and validated alternative for miR-107 quantification. The detection was performed using the Human miR-107 ELISA Kit (Shanghai Ideal Medical Technology, China), strictly following the manufacturer’s protocol. Below is the detailed procedure as outlined in the accompanying pamphlet of the kit.

''Human miR-107 ELISA Kit employs the sandwich enzyme immunoassay technique for the quantitative measurement of human miR-107 in serum, plasma, tissue homogenates, and other biological fluids. An antibody specific for miR-107 has been pre-coated onto a 96-well microtiter plate. The standards and test samples are added into the wells and the miR-107 present in each sample is bound to the wells by the immobilized antibody. Following incubation, the wells are washed and then incubated with Biotinylated Anti-miR-107 Antibody, which binds the captured miR-107 present in each well. Following incubation, unbound biotinylated detection antibody is removed by washing, and an HRP-Streptavidin conjugate is added to the wells and the microtiter plate is incubated. Following incubation and washing, TMB substrate solution is then used to visualize the HRP enzymatic reaction by catalysis to produce a blue-colored product that changes to yellow after addition of acidic stop solution. The density of yellow is proportional to the amount of miR-107 captured in each well. The concentration of miR-107 can then be calculated by reading the O.D. absorbance at 450nm in a microplate reader and referring to the standard curve.''

Comment 2: Consistency in miR-107 Notation

We appreciate the reviewer’s observation regarding the inconsistency in the notation of miR-107. In response, we have carefully reviewed the manuscript and ensured that the notation has been standardized to "miR-107" throughout the text. Thank you for bringing this to our attention.

Comment 3: Clarification of Study Conclusions

We thank the reviewer for pointing out this concern. We acknowledge the unintentional overstatement in the abstract and have revised the concluding sentence to better align with the study’s findings. We have ensured that our conclusions accurately reflect the scope and implications of our results. We appreciate the reviewer’s valuable feedback in improving the clarity and precision of our manuscript.

Comment 4: Correlation Analysis Between miR-107 and Biomarkers

We appreciate the reviewer’s comment and would like to clarify that Spearman's correlation analysis between miR-107, and the studied biomarkers has been conducted and is presented in the results section of the manuscript. Our analysis of the control samples revealed statistically significant moderate correlations between miR-107 and certain biomarkers, while no significant correlations were observed with the other studied markers. We have ensured that the significance and implications of these findings are appropriately discussed in the manuscript.

Thank you for your valuable feedback.

Comment 5: Improvement of Logical Flow in Introduction and Justification for miR-107 Selection

We appreciate the reviewer’s suggestion to enhance the logical flow of the Introduction. In response, we have restructured the section to prioritize a discussion on existing biomarkers for OSA and their limitations, followed by an explanation for the selection of miR-107.

Different biomarkers have been explored to assess the presence and severity of OSA and to evaluate its systemic effects. While biomarkers such as C-reactive protein (CRP), interleukins (IL-6, IL-8), and tumor necrosis factor-alpha (TNF-α) reflect the systemic inflammation associated with OSA and its comorbidities, they lack specificity for OSA. Similarly, oxidative stress biomarkers, including malondialdehyde (MDA), 8-isoprostane, and oxidized LDL, are associated with intermittent hypoxia and reoxygenation but suffer from a lack of standardization. Genetic biomarkers, such as TNF-α gene polymorphisms and angiotensin-converting enzyme gene polymorphisms, have been suggested to predict susceptibility to OSA or its complications. However, the complex polygenic nature of OSA, coupled with genetic variability and environmental influences, makes it challenging to identify a single definitive genetic marker. As a result, no single genetic marker is likely to provide conclusive diagnostic or prognostic information.

Given these limitations, current research is focused on identifying more precise, reliable biomarkers for OSA to complement polysomnography, the gold standard for diagnosis. In this context, miR-107 was selected as a promising biomarker due to its non-invasive nature, specificity, and direct link to the molecular events triggered by intermittent hypoxia. miR-107 has shown a strong correlation with the arousal index and sleep parameters, which are crucial in OSA. Furthermore, miR-107 regulates key pathways involving glutamate, serotonin, adiponectin, and melatonin, all of which influence cognitive function, metabolic health, and respiratory control. Dysregulation of miR-107 exacerbates glutamate toxicity, serotonin imbalance, reduced adiponectin, and disrupted melatonin production, all of which are implicated in OSA pathology.

The decision to focus on miR-107 was driven by its potential to address these interconnected issues. While further research is necessary to overcome current limitations and confirm its diagnostic and prognostic value, miR-107 stands out as one of the most promising biomarkers for advancing the clinical management of OSA.

This revision aims to enhance the logical progression of the Introduction by first addressing existing biomarkers, their limitations, and then justifying the selection of miR-107 based on its potential to overcome these challenges.

Comment 6: Correction of Figure 1

Thank you for your feedback. We included the table as Fig 1 in accordance with the journal's guidelines, which require tables presented in figure format to be referred to as figures rather than tables.

Comment 7: Discussion of BMI Differences and Influence of Obesity on miR-107 and Adiponectin

We appreciate the reviewer's insightful comments and observations. In our study, miR-107 and adiponectin levels were found to be lower in OSA patients compared to controls. The observed reductions in miR-107 and adiponectin are likely influenced by both obesity and OSA. Obesity is known to induce a state of low-grade chronic inflammation and elevate the production of pro-inflammatory cytokines, which can suppress miR-107 expression and inhibit adiponectin secretion. Furthermore, adiponectin levels are typically reduced in obese individuals, irrespective of OSA status.

In addition to obesity, OSA-specific pathophysiological mechanisms, such as intermittent hypoxia, contribute to systemic inflammation and oxidative stress, which may further downregulate miR-107 expression and impair adiponectin secretion. The recurrent episodes of airway obstruction and consequent hypoxemia exacerbate metabolic dysregulation in OSA patients, further influencing these biomarker levels.

While obesity is a significant contributing factor to these alterations, OSA-related mechanisms, including hypoxia and inflammation, also play a critical role. Ideally, a case group comprising non-obese OSA patients would provide a clearer understanding of the independent effects of OSA on these biomarkers. However, the recruitment of such individuals presents a considerable challenge, as non-obese OSA patients are relatively uncommon in clinical practice.

To mitigate the confounding effects of obesity and isolate the specific impact of OSA, we selected non-obese individuals as controls. This approach allows for a more precise evaluation of OSA-related biomarker changes, such as those driven by intermittent hypoxia and systemic inflammation, without the additional influence of obesity.

We acknowledge the reviewer’s concern regarding the selection of the control group; however, our approach was based on the need to delineate OSA-related effects from those associated with obesity, while also considering the practical limitations in recruiting non-obese OSA patients.

Final words:

We would like to inform the editor that the data used for this study has been uploaded as supporting files. We greatly appreciate your time and effort in considering our manuscript, and we would like to thank you for your valuable feedback and support throughout the review process.

Kind regards,

Dr. Kaleem Arshad

Khawaja M. Safdar Medical College, Sialkot

kaleemarshad630@gmail.com

---

## [Decision Letter · Decision Letter 1]

26 Feb 2025

PONE-D-24-51520R1Evaluating miR-107 and adiponectin as biomarkers in obstructive sleep apnea: Associations with neurotransmitters and metabolic regulationPLOS ONE

Dear Dr. Arshad,

Thank you for submitting your manuscript to PLOS ONE. Your manuscript, referenced above, has now been reviewed by experts in the field. PLOS ONE is intended to disseminate original research and research methodologies. Specifically, experiments, statistics, and other analyses should be performed to a high technical standard and should be described in sufficient detail. Additionally, experiments must have been conducted rigorously, with appropriate controls and replication. Sample sizes must be large enough to produce robust results, where applicable. After careful consideration, we feel that it has merit but does not fully meet PLOS ONE’s publication criteria as it currently stands. Therefore, we invite you to submit a revised version of the manuscript that addresses the points raised during the review process.

We look forward to receiving your revised manuscript.

Kind regards,

Assoc. Prof. Phakkharawat Sittiprapaporn, Ph.D.

Academic Editor

PLOS ONE

Journal Requirements:

Reviewers' comments:

Reviewer's Responses to Questions

**Comments to the Author**

1. If the authors have adequately addressed your comments raised in a previous round of review and you feel that this manuscript is now acceptable for publication, you may indicate that here to bypass the “Comments to the Author” section, enter your conflict of interest statement in the “Confidential to Editor” section, and submit your "Accept" recommendation.

Reviewer #1: All comments have been addressed

Reviewer #4: All comments have been addressed

2. Is the manuscript technically sound, and do the data support the conclusions?

Reviewer #1: Partly

Reviewer #4: Yes

3. Has the statistical analysis been performed appropriately and rigorously? 

Reviewer #1: Yes

Reviewer #4: Yes

4. Have the authors made all data underlying the findings in their manuscript fully available?

Reviewer #1: Yes

Reviewer #4: Yes

5. Is the manuscript presented in an intelligible fashion and written in standard English?

Reviewer #1: Yes

Reviewer #4: Yes

6. Review Comments to the Author

Reviewer #1: In this study, the difference in miR-107 levels between the OSA patients and the control persons reached approximately 66-fold (77.85 ng/mL vs. 1.18 ng/mL), far exceeding the previously reported ranges in other diseases. Is this result really correct? Is the method used in this test suitable for this kind of experiment? Is it possible to use other method such as qRT-PCR to re-evaluate the levels of miR-107?

Reviewer #4: I have reviewed the revised manuscript and find that the authors have satisfactorily addressed the concerns raised by reviewers. The incorporation of the suggested modifications has significantly enhanced the clarity and robustness of the study. Consequently, I recommend the manuscript for publication

7. PLOS authors have the option to publish the peer review history of their article (what does this mean? ). If published, this will include your full peer review and any attached files.

**Do you want your identity to be public for this peer review?** For information about this choice, including consent withdrawal, please see our Privacy Policy .

Reviewer #1: No

Reviewer #4: No

---

## [Author Response · Author response to Decision Letter 2]

27 Feb 2025

Phakkharawat Sittiprapaporn

Academic Editor

PLOS ONE

Dear Dr. Phakkharawat,

We would like to express our sincere gratitude for the opportunity to submit our manuscript titled “Evaluating miR-107 and adiponectin as biomarkers in obstructive sleep apnea: Associations with neurotransmitters and metabolic regulation” to PLOS ONE. We deeply appreciate the insightful comments provided by you and the reviewers.

We received two comments in total. One reviewer recommended the manuscript for publication, and the only point requiring our response was regarding the miR-107 estimation. As there were no concerns or suggestions for modifications to the manuscript text, no changes to the manuscript were necessary.

Regarding the reviewer’s comment on miR-107 measurement: We clarified that the miR-107 quantification was performed using the Human miR-107 ELISA Kit, strictly following the manufacturer’s instructions. Both cases and controls were assayed concurrently, and miR-107 levels were determined by comparing the optical density at 450 nm against a standard curve generated from the kit-provided standards. We are confident that this approach ensured accurate and reproducible measurements, as further supported by our simultaneous quantification of other markers using the same instruments and laboratory conditions.

Reviewer #1:

Dear Reviewer, Thank you for your insightful comment. We would like to clarify that the quantification of miR-107 was performed using the Human miR-107 ELISA Kit, and the procedure was strictly carried out according to the manufacturer's manual. Specifically, the assay employs a sandwich enzyme immunoassay technique for the quantitative measurement of human miR-107 in serum, plasma, tissue homogenates, and other biological fluids. The full method is as follows:

“Human miR-107 ELISA Kit employs the sandwich enzyme immunoassay technique for the quantitative measurement of human miR-107 in serum, plasma, tissue homogenates, and other biological fluids. An antibody specific for miR-107 has been pre-coated onto a 96-well microtiter plate. The standards and test samples are added into the wells and the miR-107 present in each sample is bound to the wells by the immobilized antibody. Following incubation, the wells are washed and then incubated with Biotinylated Anti-miR-107 Antibody, which binds the captured miR-107 present in each well. Following incubation, unbound biotinylated detection antibody is removed by washing, and an HRP-Streptavidin conjugate is added to the wells and the microtiter plate is incubated. Following incubation and washing, TMB substrate solution is then used to visualize the HRP enzymatic reaction by catalysis to produce a blue-colored product that changes to yellow after addition of acidic stop solution. The density of yellow is proportional to the amount of miR-107 captured in each well. The concentration of miR-107 can then be calculated by reading the O.D. absorbance at 450nm in a microplate reader and referring to the standard curve. Both cases and controls were assayed simultaneously using the same batch of reagents, instruments, and standardized protocols, ensuring minimal inter-assay variability. We also quantified other biomarkers in parallel using the same instrumentation and laboratory conditions, and those measurements confirmed the overall consistency and reliability of our procedures.”

While we acknowledge that our results indicate a wide difference in miR-107 levels between OSA patients and controls, literature does report significant differences in miRNA expression across groups, albeit not always as pronounced as in our study. It is important to note that at the time of our study, we faced constraints including limited access to a qRT-PCR facility and budget restrictions. Consequently, we opted for the ELISA method, which was already in use for the quantification of other markers in our laboratory. Moreover, as the participants were recruited last year, the available serum samples have been fully utilized, and the current funding does not allow for additional qRT-PCR analyses.

We are confident in the accuracy and validity of our results, as the procedures were rigorously standardized and executed, ensuring reliable data. We appreciate your valuable feedback and hope this response adequately addresses your concerns.

Reviewer #4:

Thank you very much for your positive feedback and for recommending our manuscript for publication. We truly appreciate the time and effort you invested in reviewing our work. Your constructive comments and recognition of the enhancements made to the clarity and robustness of our study have been invaluable to us. We are delighted that the modifications have satisfactorily addressed the concerns raised and look forward to the next steps.

Final Words:

Thank you again for the opportunity to submit our work and for your valuable feedback. We hope our responses have adequately addressed the comments and look forward to your favorable decision on the acceptance of our manuscript.

Kind regards,

Dr. Kaleem Arshad

Khawaja M. Safdar Medical College, Sialkot

kaleemarshad630@gmail.com

---

## [Decision Letter · Decision Letter 2]

17 Mar 2025

Evaluating miR-107 and adiponectin as biomarkers in obstructive sleep apnea: Associations with neurotransmitters and metabolic regulation

PONE-D-24-51520R2

Dear Dr. Arshad,

We’re pleased to inform you that your manuscript has been judged scientifically suitable for publication and will be formally accepted for publication once it meets all outstanding technical requirements.

Kind regards,

Assoc. Prof. Phakkharawat Sittiprapaporn, Ph.D.

Academic Editor

PLOS ONE

Reviewers' comments:

Reviewer's Responses to Questions

**Comments to the Author**

1. If the authors have adequately addressed your comments raised in a previous round of review and you feel that this manuscript is now acceptable for publication, you may indicate that here to bypass the “Comments to the Author” section, enter your conflict of interest statement in the “Confidential to Editor” section, and submit your "Accept" recommendation.

Reviewer #1: All comments have been addressed

2. Is the manuscript technically sound, and do the data support the conclusions?

Reviewer #1: Yes

3. Has the statistical analysis been performed appropriately and rigorously? 

Reviewer #1: Yes

4. Have the authors made all data underlying the findings in their manuscript fully available?

Reviewer #1: Yes

5. Is the manuscript presented in an intelligible fashion and written in standard English?

Reviewer #1: Yes

6. Review Comments to the Author

Reviewer #1: I believe that the authors had tried their best to give us some meaningful data to help the diagonosis and treament of the obstructive sleep apnea patients. So I suggest the publicaiton of this paper in PloS One. And I hope the authors will do better in future.

7. PLOS authors have the option to publish the peer review history of their article (what does this mean? ). If published, this will include your full peer review and any attached files.

**Do you want your identity to be public for this peer review?** For information about this choice, including consent withdrawal, please see our Privacy Policy .

Reviewer #1: No

---

## [Editor Report · Acceptance letter]

PONE-D-24-51520R2

PLOS ONE

Dear Dr. Arshad,

I'm pleased to inform you that your manuscript has been deemed suitable for publication in PLOS ONE. Congratulations! Your manuscript is now being handed over to our production team.

Kind regards,

on behalf of

Assoc. Prof. Dr. Phakkharawat Sittiprapaporn

Academic Editor

PLOS ONE